# Inferior Vena Cava Ultrasonography for Volume Status Evaluation: An Intriguing Promise Never Fulfilled

**DOI:** 10.3390/jcm12062217

**Published:** 2023-03-13

**Authors:** Pierpaolo Di Nicolò, Guido Tavazzi, Luca Nannoni, Francesco Corradi

**Affiliations:** 1Nephrology and Dialysis Unit, S. Maria della Scaletta Hospital, 40026 Imola, Italy; 2Department of Clinical Surgical, Diagnostic and Pediatric Sciences, University of Pavia, 27100 Pavia, Italy; 3Anaesthesia, Intensive Care and Pain Therapy, Fondazione IRCCS Policlinico San Matteo, 27100 Pavia, Italy; 4Radiology Unit, S. Maria della Scaletta Hospital, 40026 Imola, Italy; 5Department of Surgical, Medical, Molecular Pathology and Critical Care Medicine, University of Pisa, 56124 Pisa, Italy; 6Azienda Ospedaliero Universitaria Pisana Via Paradisa, 2, 56126 Pisa, Italy

**Keywords:** inferior vena cava ultrasonography, volume status, central venous pressure, fluid responsiveness, collapsibility index, distensibility index

## Abstract

The correct determination of volume status is a fundamental component of clinical evaluation as both hypovolaemia (with hypoperfusion) and hypervolaemia (with fluid overload) increase morbidity and mortality in critically ill patients. As inferior vena cava (IVC) accounts for two-thirds of systemic venous return, it has been proposed as a marker of volaemic status by indirect assessment of central venous pressure or fluid responsiveness. Although ultrasonographic evaluation of IVC is relatively easy to perform, correct interpretation of the results may not be that simple and multiple pitfalls hamper its wider application in the clinical setting. In the present review, the basic elements of the pathophysiology of IVC behaviour, potential applications and limitations of its evaluation are discussed.

## 1. Introduction

Correct determination of the volume status of the patient represents a fundamental step in clinical evaluation. In fact, in many critically ill patients hypovolaemia might result in reduced tissue perfusion, while fluid overload can lead to organ congestion with associated morbidity and mortality [1]. Thus, volaemic assessment is essential to guide clinicians in treatment and might have a potential prognostic implication.

Right atrial pressure (RAP) is a cornerstone of evaluation of the intravascular volume status that predicts overall survival in patients with heart failure [2]; the terms ‘‘central venous pressure’’ (CVP) and ‘‘right atrial pressure’’ (RAP) are synonymous as long as there is no vena cava obstruction, and they will be used interchangeably in this manuscript.

The standard monitoring tool for assessing CVP is the central venous catheter. However, due to its invasiveness, infectious and thrombotic complications are considered major concerns. In this context, point-of-care ultrasonography (POCUS) might be a reliable alternative for volume estimation.

Ultrasonographic evaluation of the IVC (US-IVC) has been proposed as the non-invasive technique of choice for CVP assessment due to its wide availability, low costs, and ease of use. Estimation of volume status by measuring the static diameter of the IVC and dynamic respiratory fluctuations has been investigated in various clinical settings; nevertheless, results are often conflicting and uncertain, as hypervolaemia and hypovolaemia are not the only factors affecting IVC diameter. Hence, although US-IVC is relatively easy to apply, its use in clinical decision-making requires a deep knowledge of pathophysiology, limitations and pitfalls to avoid perceptual and interpretive errors.

The aim of this paper is thus to review the main physiopathology about venous return and right atrial filling, explore the current evidence (accuracy and pitfalls) of US-IVC in clinical practice, and give clinicians the instruments for its correct interpretation.

## 2. Pathophysiological Assumptions

The IVC has the largest diameter of the entire venous system; it is a thin-walled, valveless, retroperitoneal vessel, responsible for returning large volumes of deoxygenated blood from the lower extremities and abdomen to the right atrium. With 85% of total plasma volume in the venous circulation, the IVC is an important blood reservoir, and modifications of circulating volume result in IVC calibre variations.

Indeed, evidence of a ‘’flat vena cava” (e.g., an IVC with an anteroposterior diameter of less than 9 mm) at multiple levels is associated with significant hypovolaemia in trauma patients [3].

Patient position and decubitus can influence circulating blood volume and IVC diameter by gravity: the IVC is smaller when the patient is in the left lateral position and larger when the patient is in the right lateral position [4]. 

In addition to circulating volume, other important factors can lead to variations in IVC diameter during the respiratory cycle, such as right heart function and the gradient between intrathoracic and intra-abdominal pressure (Figure 1).

### 2.1. Intrathoracic Pressure

In spontaneously breathing subjects, during inspiration abdominal pressure increases while intrathoracic pressure (ITP) increases its negativity, lowering right atrial pressure. The haemodynamic counterpart to the aforementioned thoraco–abdominal interaction is the increase in blood return from the IVC to the RA, leading to a secondary reduction in the size of the IVC [5] and a transient increase in stroke volume. Conversely, venous return decreases during exhalation, while the calibre of the IVC increases [6,7]. 

IVC collapsibility may be exaggerated if ITP becomes markedly negative with forced inspiratory efforts, respiratory distress, or exacerbation of chronic obstructive pulmonary disease, causing an increased venous return to the right atrium.

In ventilated patients, positive end-expiratory pressure (PEEP) may hamper the venous return during inspiration by increasing the ITP and reducing the pressure gradient between the abdominal and thoracic compartments [8]. This issue may be critical in cases of pre-load dependence (e.g., right ventricular infarction, pulmonary embolism, tamponade, severe hypovolaemia), leading to the abrupt reduction in venous return that triggers haemodynamic instability [9].

### 2.2. Intra-Abdominal Pressure

Intra-abdominal pressure (IAP) may influence IVC physiology even more than ITP, as it affects both venous return and IVC diameter. In the subdiaphragmatic region, when the transmural pressure of the IVC exceeds the critical closing pressure, the IVC is pervious; in this situation, an increase in abdominal pressure reduces IVC diameter and increases transitorily the venous return, with the liver serving as the immediate blood source [7]. Conversely, when the IVC pressure is below the critical closing pressure, the increased intrabdominal pressure causes the IVC collapse with a dramatic drop in venous return [10]. Therefore, increased intra-abdominal pressure during inspiration might have opposite effects on total and regional IVC venous return.

Not only IAP but also the volaemic status can influence venous return during the respiratory cycle. In case of hypervolaemia, the active diaphragmatic descent causes a significant increase in total IVC flow by enhancing splanchnic venous return through the IVC. On the other hand, in case of hypovolaemia the possibly increased abdominal pressure reduces venous return, leading to a decrease in IVC flow at the level of the diaphragm (e.g., vascular waterfall effect) [11].

Therefore, two important factors may affect IVC flow and diameter and thus venous return: intra-abdominal pressure and volaemia. Regardless of the volaemic state, severe abdominal hypertension always causes a drop in the IVC venous return and a consequent decrease in cardiac output [12].

### 2.3. Cardiac Function

Because of the close relationship between venous return and right atrial pressure, the wall movements of the IVC reflect the haemodynamic behaviour of the right atrium (RA) under both physiological and pathological conditions [13], which are influenced by the cardiac cycle, right heart function, and rhythm.

In sinus rhythm, the IVC has its maximum diameter during the presystolic and systolic phases [5] while atrial fibrillation alters the filling of the IVC, making the relationship between cardiac cycle and IVC dimension difficult to assess. IVC diameter should be interpreted in the light of the physiology of venous return, right heart function and heart-lung interaction.

Indeed, the inspiratory collapsibility of the IVC with normal ITP is an expression of the adequacy of the right heart to reduce RAP (Figure 2B) [14]. In response to increased ITP, the curvilinear relationship between IVC diameter and CVP is clearly evident (Figure 2A), with an initial steep part (i.e., a minimal increase in CVP determines a large increase in IVC diameter) and a flat part (a larger increase in CVP causes minimal or no IVC dilation [9].

Under pathological conditions, such as acute circulatory failure, the change in IVC diameter is thus a function of residual venous compliance and cardiac functional reserve (Figure 2A,B).

Therefore, the behaviour of the IVC is the result of a complex interplay between the heart, volaemia, and respiratory mechanics acting simultaneously in different clinical contexts (Figure 1).

## 3. Anatomical Variations of Inferior Vena Cava and Their Clinical Significance

Due to a complex embryogenesis process, congenital anomalies of the IVC and its tributaries are not uncommon, with a reported prevalence of 0.3% to 10.14% of the population and a total of 14 observed variations; knowledge of these clinical entities may avoid severe consequences during retroperitoneal surgical procedures (especially laparoscopic procedures) [16,17].

The most common IVC anomalies are duplication, with a left IVC terminating below or at the level of the left renal vein, a left-sided IVC, and interruption or agenesia of the IVC [18,19].

From a clinical point of view, IVC duplication together with left-sided vena cava and megacava are generally asymptomatic, whereas aplasia and hypoplasia may be associated with iliofemoral deep venous thrombosis [16]. 

In all these conditions, both the calibre of the vessel (or vessels), the venous return and the behaviour of the IVC in relation to various volaemic conditions may be severely altered.

## 4. Ultrasound Technique, Static and Dynamic IVC Indexes 

Either a low-frequency convex probe (2–5.5 MHz) for the abdomen or a phased array transducer (2–8 MHz) for the heart can be used to assess the IVC.

The IVC is usually visualized from a subcostal view by a longitudinal scan, including the veno–atrial junction and the right atrium with inner walls clearly visible. In case of a suboptimal or unavailable subcostal window, a coronal transhepatic scan along the posterior right axillary line may be an effective alternative (Figure 3).

The exact position at which the IVC diameter should be measured is crucial, although it is not universally standardized: in spontaneously breathing patients, variations in the IVC diameter are smaller near the right atrium and larger 2 cm caudal to the hepatic vein inlet or at the level of the left renal vein [20]. Most authors suggest that measurements should be acquired within 1.5 cm from the IVC-to-right atrial junction [21]. 

B-mode is used for the identification of the IVC and then the M-mode Doppler is applied with a sweep velocity set at 25 to 50 mm/s, depending on the respiratory rate, to capture at least three respiratory cycles.

Since the minimum venous diameter in spontaneously breathing patients may be influenced by inspiratory effort, maximal inspiration (sniffing manoeuvre) could be evoked while maximal IVC diameter is measured at the end of expiration [22].

Sample accuracy can be improved by using indexed IVC size (iIVC), which is calculated by dividing IVCmax by body surface area [23].

In addition to these static parameters, it is always useful to make a dynamic assessment using the IVC collapsibility index (cIVC), which is calculated according to the following formula: cIVC = (IVCmax − IVCmin)/IVCmax [24,25]. 

In mechanically ventilated patients, the IVC distensibility index, described as

dIVC = (IVCmax − IVCmin)/IVCmin, [21] the respiratory variations in inferior vena cava diameter as ΔDIVC = (IVCmax − IVCmin)/(IVCmax + IVC min)/2) [26] or the IVC Area Distensibility Index (IVC-ADI) defined as maximum IVC area-minimum IVC area)/minimum IVC area × 100% can be applied [27]. 

Echocolour and pulsed wave Doppler assessment of the venous spectrometric wave, although not commonly performed in IVC, can be helpful in certain clinical contexts such as stenosis/thrombosis or congenital anomalies [28,29].

The technical limitations of US-IVC are obesity or pregnancy, chest or gastric tubes, and non-negligible inter/intra-observer variability often caused by lateral displacement of the IVC during respiration, so that both the true centre of the vein and the accuracy of the measurement are lost in M-mode imaging [30,31].

## 5. Evidence for Volaemic Status Evaluation with IVC Ultrasonography

IVC diameter and its temporal changes in the respiratory cycle have long been studied to correlate with CVP and predict fluid responsiveness.

These two aspects are addressed here separately according to the pathophysiological premises.

### 5.1. Volaemic Status Evaluation in Spontaneously Breathing Patient

Overall, a statistically significant non-linear correlation was described between the sonographic dimensional parameters of IVC and CVP [32]. Most studies demonstrated a moderate correlation between measurements of IVC diameter or collapsibility and CVP or RAP [33]. Cut-off values of 2 cm diameter and cIVC of 40% provided the best diagnostic accuracy in predicting a RAP above or below 10 mmHg [34,35,36].

According to the current updated American and European guidelines, an IVC diameter ≤2.1 cm and collapsibility >50% during inspiration suggest a RAP between 0–5 mm Hg while a diameter >2.1 cm with <50% inspiratory collapse indicates a high RAP of 10–20 mmHg; a mean pressure value of 8 mmHg is used if the clinical picture does not follow the proposed pattern [37].

In outpatients undergoing haemodialysis, standardization of IVC diameter to body surface area (BSA) was recommended (i.e., IVC diameter 2.1 cm if BSA > 1.61 m^2^, IVC diameter 1.7 cm when BSA < 1.61 m^2^) [38] and an indexed IVC size (*i*IVC) ≥ 8 and ≤11.5 mm/m^2^ is considered safe to rule out severe hyper or hypovolaemia in this setting [23,36].

The addition of pulsed wave Doppler applied to the IVC may provide additional information to estimate CVP (Figure 3). The presence of continuous flow from the IVC to the RA corresponds to a low to normal CVP; on the contrary, an interrupted waveform indicates a high RAP only if it is associated with other ultrasound indices such as the IVC size and cIVC [29].

It has also been described that the IVC diameter and cIVC correlate with plasma volume removal by ultrafiltration in continuous and intermittent haemodialysis or blood donation [39,40].

For the assessment of fluid responsiveness, the US-IVC diameters are useless [41,42]. Hence “dynamic” measurements have been developed to predict the response to volume infusion and guide fluid resuscitation (Table 1). To this end, cIVC measurements are taken before intravenous fluid administration or passive leg raising and then the cardiac output response is observed.

From all these studies it emerged that only extreme cIVC values (i.e., a cIVC value > 40%) may be useful in predicting the haemodynamic response to volume expansion [30,43], due to multiple pitfalls that can affect respiratory mechanics and cardiopulmonary interactions.

Considering that even a reduced cIVC may not rule out a fluid responsiveness or the need for fluid therapy, a cIVC threshold < 15% with the addition of a standardized inspiratory effort was proposed as an attempt to improve cIVC diagnostic accuracy [44].

Therefore, in spontaneously breathing patients, a cIVC-guided fluid infusion can be considered a logical, but rarely decisive, option before the administration of infusion therapy.

### 5.2. Volaemic Status Evaluation in Mechanically Ventilated Patients 

Regarding the non-invasive estimation of CVP, current evidence does not support the use of the IVC diameter in mechanically ventilated patients. In a recent meta-analysis that included 16 studies, the correlation between CVP and IVC diameter was not statistically significant in 8 studies and was weak to moderate in the others [33]. These results may be due to the complex interplay between intrathoracic pressure, right atrial pressure and venous return, so that a unique interpretation can be challenging in most cases (Figure 1) [37].

Evaluation of fluid responsiveness relies on the IVC’s potential to dilate by increasing its diameter during positive pressure ventilation, shifting from the steep to the flat part of the IVC-to-CVP curve (Figure 2A) [9,15]. However, in 540 subjects with acute circulatory failure of any cause (the largest adult population ever studied on this topic), respiratory variations in the IVC diameter provided only weak to moderate diagnostic accuracy in predicting fluid responsiveness; this result was likely due to concurrent abdominal hypertension and/or low level of mechanical power during protective mechanical ventilation (e.g., low tidal volume of less than 8 mL/kg predicted body weight, moderate to low positive end expiratory pressure, low respiratory rate, low driving pressures) [45].

Regarding the paediatric population, the evidence for patients on mechanical ventilation is quite sparse and contrasting. In 21 children who had undergone cardiac surgery, the ΔD_IVC_ accurately predicted fluid responsiveness, whereas it did not in 33 neurosurgical patients [46,47].

Likewise, in the work by Weber et al. on 31 subjects (aged 1 day to 13 years), respiratory cycle-induced changes in IVC diameter were useless for predicting fluid responsiveness [48,61]. These results may be explained by the higher IVC elasticity and chest wall compliance in the paediatric population compared with the adults, resulting in dampened transpulmonary pressure [48].

Regardless, we must note that in most of the above-mentioned adult and paediatric studies, cardiac function data were not available, which is a significant limitation in the assessment reliability of IVC US.

In mechanically ventilated subjects, a new distensibility index based on IVC area was recently proposed by Yao and coworkers (VCI ADI, cut-off value 10.2%), which revealed higher sensitivity in predicting fluid responsiveness than dIVC, even though it was loaded with very low specificity (97.3% sensitivity and 40.0% specificity, respectively) [23].

In summary, for fluid responsiveness assessment, dIVC has a better diagnostic performance than cIVC in spontaneously breathing patients [26]; its clinical utility in patients receiving mechanical ventilation is questionable and can be only applied in the context of a preserved biventricular heart function. Moreover, dIVC is not adequately supported by the currently available evidence [62] in abdominal surgery, concurrent abdominal hypertension, patients ventilated with protective mechanical ventilation [63] and the paediatric population, as it is burdened by poor diagnostic accuracy.

## 6. Pitfalls That May Lead to Misinterpretation of Volume Status

IVC dilatation had been suggested as a potential predictor of outcome in many different clinical contexts [64,65,66]. However, numerous physiological and pathophysiological conditions, as well as several interpretive pitfalls, have limited its wide applicability [24].

IVC diameter interpretation in relation to volume status includes several confounding conditions, such as: chronically dilated IVC in young trained athletes due to adaptation to chronic strenuous exercise, young patients with vasovagal syncope that present increased venous pooling, children with increased venous compliance, direct vasoplegic effect of drugs or sepsis, [67,68] severe tricuspid regurgitation, right ventricular failure, pericardial effusion or tamponade, acute pulmonary embolism, intra-abdominal hypertension including pregnancy, [69,70,71] (Figure 1), COPD exacerbations with hyperinflation and increased intrathoracic pressure [72], and patients undergoing mechanical ventilation with high positive end-expiratory pressure [73]. Similarly, local mechanical factors such as IVC stenosis and thrombosis or the presence of devices such as cava filters and catheters can reduce venous return. In all the above-mentioned conditions, chronically strained IVC can be dilated without an underlying hypervolaemic state. 

On the contrary, the presence of masses compressing the IVC and COPD exacerbations with forced expiration may mimic IVC collapsibility [71]; when circulating volume depletion occurs due to severe hypoproteinaemia such as in liver cirrhosis, malnutrition, cancer or sepsis, the IVC reduces in size, but the patient can be hyper-hydrated due to the massive shift of fluid into the third space, leading to splanchnic congestion regardless of the behaviour of the IVC.

## 7. IVC Ultrasonography: Current Knowledge and Future Directions

The reliability of IVC US as an index of volaemic status varies greatly in different clinical contexts and this aspect must be considered before drawing therapeutic conclusions.

For all the above reasons, and because of its complex pathophysiology, the inferior vena cava evaluation has numerous interpretive pitfalls that make it useless for inferring volume status when considered as a sole measure. Recently, a combined evaluation of IVC and other body districts’ veins has been proposed to minimize its interpretative drawbacks and have a more comprehensive view of the congestion [74]. However, even in this case, IVC-US cannot be considered as a gateway to decide whether to proceed with the investigation of the splanchnic compartment, as splanchnic congestion can be present independently from IVC’s behaviour. Even if it is time-consuming, only a combined US assessment of IVC, heart, lung, portal, splenic and renal veins, (Figure 4) may provide additional insights to explain a complex pathophysiology [75].

## 8. Conclusions

Assessment of volume status is a cornerstone of clinical evaluation, and for this purpose we cannot ignore the major advantages of IVC ultrasonography such as non-invasivity, wide availability, low cost, relative ease of use and repeatability.

On the other hand, its use is burdened by technical limitations, errors in interpretation and limited areas of clinical application. 

For these reasons, IVC-US cannot be considered a stand-alone method suitable for all patients, and a comprehensive assessment of organ perfusion in the critically ill patients requires a clinical physiopathological perspective in conjunction with an integrated ultrasonographic approach, combining multiple sites of investigation.

## Figures and Tables

**Figure 1 jcm-12-02217-f001:**
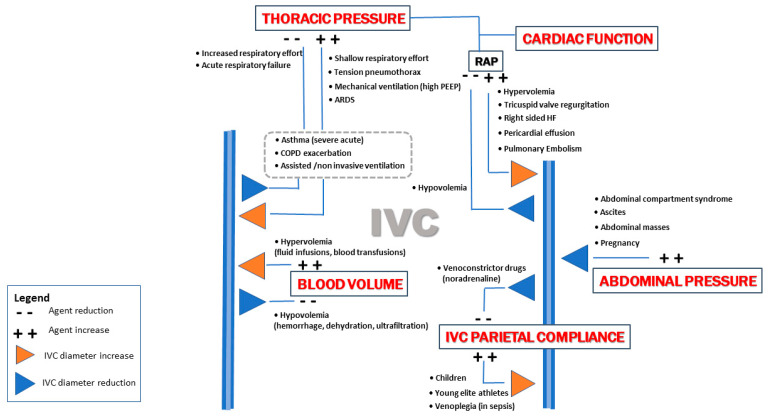
Main determinants affecting inferior vena cava diameter. Chest pressure can act directly and/or indirectly (via the RAP) on IVC diameter. The clinical conditions within the grey dashed rectangle can correlate with both types of chest pressure variations. (ARDS: acute respiratory distress syndrome, COPD: chronic obstructive pulmonary disease, HF: heart failure, IVC: inferior vena cava, PEEP: positive end-expiratory pressure, RAP: right atrial pressure).

**Figure 2 jcm-12-02217-f002:**
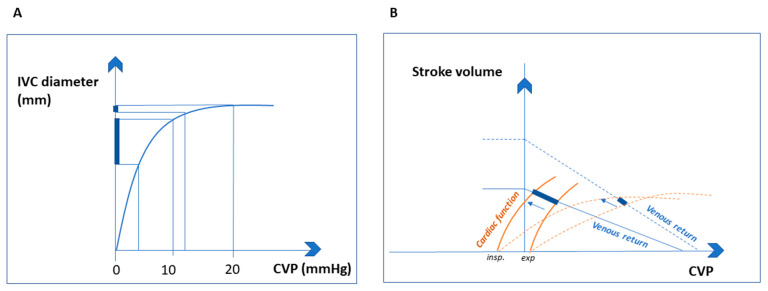
Inferior vena cava diameter as a function of residual venous compliance (**A**). In the initial ascending part, a small variation in CVP significantly increases IVC diameter. In the second part, IVC compliance decreases and a larger increase in CVP causes minimal IVC dilation. Modified from [9,15]. Inferior vena cava diameter as a function of cardiac functional reserve (**B**). The intersection between the venous return and cardiac function curves is shown for subjects with normal (solid lines) and impaired cardiac function (dotted lines). Only when cardiac function is preserved can inspiration shift the cardiac function curve to the left with CVP reduction and IVC collapse. Modified from [14]. (CVP: central venous pressure, IVC: inferior vena cava).

**Figure 3 jcm-12-02217-f003:**
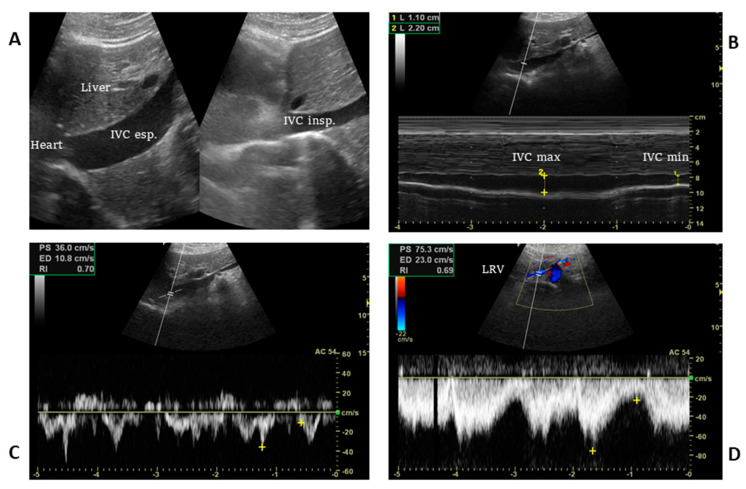
A longitudinal scan of the inferior vena cava including the veno–atrial junction (**A**) and the right coronal transhepatic scan along the posterior right axillary line (**B**). B-mode (**A**) is used to identify the inferior vena cava and then the Doppler M-mode (**B**) is applied with the sweep velocity set at 25 to 50 mm/s depending on the respiratory rate in order to include at least three respiratory cycles. The maximum and minimum IVC diameters are used to obtain the collapsibility index (in the example, cIVC is 50%). Pulsed wave Doppler in the IVC (**C**) and at the outlet of the left renal vein (**D**) may provide additional information to estimate CVP, as the presence of continuous flow equates to low to normal central venous pressure. (IVC: inferior vena cava, LRV: left renal vein, yellow + : peak velocity at end-expiration and end-inspiration).

**Figure 4 jcm-12-02217-f004:**
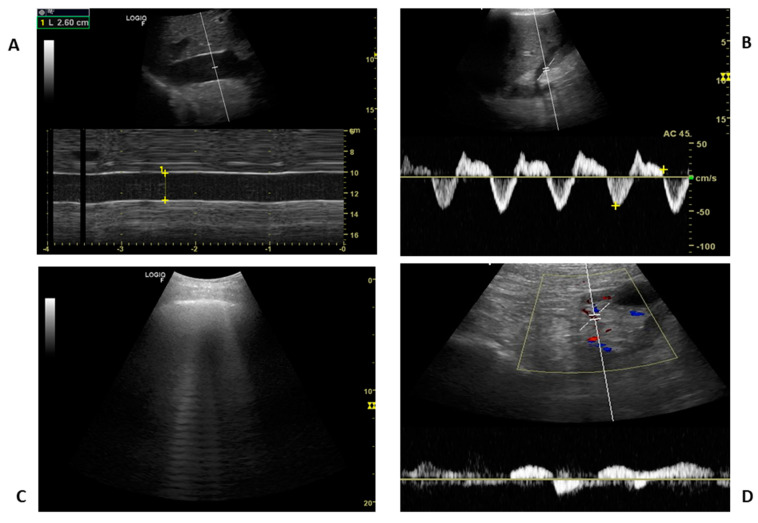
A combined approach of IVC (**A**), suprahepatic veins (**B**), lung (**C**) and renal veins (**D**) allows a more comprehensive assessment of the extent and severity of systemic venous congestion. Note that the IVC appears dilated and almost motionless during breaths (IVC max 2.6 cm, cIVC close to 0%).

**Table 1 jcm-12-02217-t001:** Fluid responsiveness in spontaneous breathing or mechanical ventilated patients.

Study	Patients (n)	Setting	Parameter	Cut-Off (%)	S (%)	Sp (%)
Spontaneously breathing patients						
Airapetian et al. [42]	59	Hypovolaemia	cIVC	≥42	31	97
Lanspa et al. [43]	14	Sepsis	cIVC	≥50	NR	NR
Bortolotti et al. [44]	55	Sepsis	cIVC-st	≥39	93	88
Vignon et al. [45]	422	Shock	dIVC	>18	28	90
Byon et al. [46]	33	Paediatric neurosurgery	ΔD_IVC_	NR	NR	NR
Choi et al. [47]	21	Paediatric cardiac surgery	ΔD_IVC_	NR	NR	NR
Weber et al. [48]	31	Paediatric	dIVC	NR	NR	NR
Muller et al. [49]	40	Shock	cIVC	>40	70	80
Preau et al. [50]	90	Sepsis	cIVC-st	≥48	84	90
Corl et al. [51]	124	Shock	cIVC	≥25	87	81
Doucet et al. [52]	144	Trauma	cIVC	≥51	NR	NR
Machare-Delgado [53]	25	Shock	dIVC	>12	100	53
Charbonneau et al. [54]	44	Sepsis	dIVC	>21	38	61
Theerawit et al. [55]	29	Sepsis	dIVC	≥10	75	77
Lu et al. [56]	49	Sepsis	dIVC	≥20	67	77
Zhang et al. [57]	40	Elective GI surgery	dIVC	≥46	69	93
Sobczyk et al. [58]	50	Cardiac Surgery	dIVC	> 18	NR	NR
Sobczyk et al. [59]	35	Cardiac Surgery	dIVC	≥ 18	82	73
Moretti et al. [60]	29	SAH	dIVC	>16	70	100
De Valk et al. [17]	45	Shock	cIVC	≥36.5	83	67
Long et al. [30]	291	Meta-analysis	cIVC	>42	52	77
Mechanical Ventilated patients						
Barbier et al. [15]	20	Sepsis	dIVC	>18	90	90
Feissel et al. [26]	39	Sepsis	dIVC	>12	NR	NR
Yao et al. [27]	67	Mixed	IVC ADI	≥10.2	97	40
			dIVC	≥25.5	46	90
Long et al. [30]	242	Meta-analysis	dIVC	>16	67	68

Table legend: NR: not reported; n: numbers; S: Sensitivity; Sp: specificity; cIVC: inferior vena cava collapsibility index; cIVC-st; dIVC: inferior vena cava distensibility index; ΔD_IVC_: respiratory variation in IVC diameter; IVC ADI: IVC Area Distensibility Index; SAH: subarachnoid haemorrhage.

## Data Availability

Not applicable.

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
