# Peer review of "Inferior Vena Cava Ultrasonography for Volume Status Evaluation: An Intriguing Promise Never Fulfilled"

_jcm, 2023, doi:10.3390/jcm12062217_

Round 1

Reviewer 1 Report

Thank you for possibility of reviewing the manuscript "Inferior Vena Cava ultrasonography for volume status evaluation: an intriguing promise never fulfilled ". It is interesting review including literature analysis focus of ultrasonographic investigation of inferior vena cava. Manuscript is potentially interesting. However, represented simply description do not associate with any statistical analysis. I recommend revision according to my suggestions. I hope my suggestion should improve the paper. Additionally it worth mention the manuscript is also important from clinical point of view.

1. Add to manuscript chapter describing most common anatomical variations of inferior vena cava (for example duplication). Base on recent literature describe volume status in such situations? Why is it so important in clinical practice?

2. Describe limitation and factors that directly and indirectly can influence on ultrasonography for volume status evaluation.

3. More clearly summarize the advantages and disadvantages of presented in manuscript method (as independent chapter).

4. Please change "conclusion" section. It is too short and include only one sentence. Why do you put citation in this place (as [59]") ? . Conclusion should base on your  study and thoughts 5. Figure 1 - explain in figures legend all abbreviation (like RAP etc. ) .

6. Manuscript language revision needed.

Author Response

  1. Add to manuscript chapter describing most common anatomical variations of inferior vena cava (for example duplication). Base on recent literature describe volume status in such situations? Why is it so important in clinical practice?

       According to reviewer’s suggestions a paragraph on IVC abnomalities and their clinical significance (particularly with regard to the assessment of fluid status) was added to the manuscript.

  1. Describe limitation and factors that directly and indirectly can influence on ultrasonography for volume status

       The chapter “Pitfalls that may lead to misinterpretation interpretation of volume status” has been expanded to include other factors that were not previously considered in the text, in order to provide a more comprehensive description of all aspects involved in volume status determination

  1. More clearly summarize the advantages and disadvantages of presented in manuscript method (as independent chapter).

We have expanded the chapter “Conclusions” to more clearly summarize the advantages and disadvantages of the IVC ultrasound examination already presented in the text.

  1. Please change "conclusions" section. It is too short and include only one sentence. Why do you put citation in this place (as [59]") ? . Conclusion should base on your study and thoughts5. Figure 1 - explain in figures legend all abbreviation (like RAP etc).

The “conclusions” chapter has been rewritten according to the reviewer’s suggestion.

The quotation has been removed from the conclusions paragraph.

The legends of the figures have been rewritten with a legend for the abbreviations

  1. Manuscript language revision needed

The manuscript has been completely revaluated and corrected by a native English speaker.  

Reviewer 2 Report

Di Nicolò and colleagues summarize the current evidence on the use of IVC ultrasonography to evaluate volume status.

I have no content-related criticism, as I completely agree with the conclusion that IVC sonography is only a tiny piece of the whole picture to judge about volume status. However, the text contains many one-sentence paragraphs making it hard to read. Please join related sentences to clarify your structure in argumentation. Furthermore, the text contains some typos that should be resolved prior to publication.

Overall, I congratulate the authors for this comprehensive review of the topic.

Author Response

  1. The text contains many one-sentence paragraphs making it hard to read. Please join related sentences to clarify your structure in argumentation. Furthermore, the text contains some typos that should be resolved prior to publication.

The manuscript has been completely revised and many one-sentence parts have been combined into a more fluent text. The typos have been corrected.

Round 2

Reviewer 1 Report

-